## [Decision Letter · Decision Letter 0]

Dear Dr. Czerwinski,

Thank you for submitting your manuscript to PLOS ONE. After careful consideration, we feel that it has merit but does not fully meet PLOS ONE’s publication criteria as it currently stands. Therefore, we invite you to submit a revised version of the manuscript that addresses the points raised during the review process.

We look forward to receiving your revised manuscript.

Kind regards,

Rong-San Jiang

Academic Editor

PLOS ONE

Journal Requirements:

2. We note that the original protocol file you uploaded contains a confidentiality notice indicating that the protocol may not be shared publicly or be published. Please note, however, that the PLOS Editorial Policy requires that the original protocol be published alongside your manuscript in the event of acceptance. Please note that should your paper be accepted, all content including the protocol will be published under the Creative Commons Attribution (CC BY) 4.0 license, which means that it will be freely available online, and any third party is permitted to access, download, copy, distribute, and use these materials in any way, even commercially, with proper attribution.

Therefore, we ask that you please seek permission from the study sponsor or body imposing the restriction on sharing this document to publish this protocol under CC BY 4.0 if your work is accepted. We kindly ask that you upload a formal statement signed by an institutional representative clarifying whether you will be able to comply with this policy. Additionally, please upload a clean copy of the protocol with the confidentiality notice (and any copyrighted institutional logos or signatures) removed.

3. We noted there is an error in the clinical trial registration number in the manuscript. Please provide the correct registration number [NCT05637307] in the manuscript.

4. In the online submission form you indicate that your data is not available for proprietary reasons and have provided a contact point for accessing this data. Please note that your current contact point is a co-author on this manuscript. According to our Data Policy, the contact point must not be an author on the manuscript and must be an institutional contact, ideally not an individual. Please revise your data statement to a non-author institutional point of contact, such as a data access or ethics committee, and send this to us via return email. Please also include contact information for the third party organization, and please include the full citation of where the data can be found.

5. Please amend either the title on the online submission form (via Edit Submission) or the title in the manuscript so that they are identical.

7.We note that Figure 2 includes an image of a patient in the study.

As per the PLOS ONE policy (http://journals.plos.org/plosone/s/submission-guidelines#loc-human-subjects-research ) on papers that include identifying, or potentially identifying, information, the individual(s) or parent(s)/guardian(s) must be informed of the terms of the PLOS open-access (CC-BY) license and provide specific permission for publication of these details under the terms of this license. Please download the Consent Form for Publication in a PLOS Journal (http://journals.plos.org/plosone/s/file?id=8ce6/plos-consent-form-english.pdf ). The signed consent form should not be submitted with the manuscript, but should be securely filed in the individual's case notes. Please amend the methods section and ethics statement of the manuscript to explicitly state that the patient/participant has provided consent for publication: “The individual in this manuscript has given written informed consent (as outlined in PLOS consent form) to publish these case details”.

Additional Editor Comments:

Please revise your manuscript based on the reviewers' comments.

Reviewers' comments:

Reviewer's Responses to Questions

**Comments to the Author**

1. Is the manuscript technically sound, and do the data support the conclusions?

Reviewer #1: Yes

Reviewer #2: Yes

2. Has the statistical analysis been performed appropriately and rigorously?

Reviewer #1: N/A

Reviewer #2: No

3. Have the authors made all data underlying the findings in their manuscript fully available?

Reviewer #1: Yes

Reviewer #2: Yes

4. Is the manuscript presented in an intelligible fashion and written in standard English?

Reviewer #1: Yes

Reviewer #2: Yes

Reviewer #1: The manuscript, titled “Sentinel NOSE: prospective feasibility study on sentinel lymph node biopsy in bulky nasal vestibule cancer” is a very well written paper, containing important information on a clinically relevant topic. In my opinion, the study should be published; however, before publishing some minor revisions should be applied.

1. The cut-off of 1.5 cm for the indication of SLNB is not sufficiently explained.

2. In the first paragraph of the introduction, the authors refer to studies which describe the nodal recurrence of NVC; however, the present study investigate how to improve accurate primary staging. I understand that there are no recent studies available on the incidence of occult lymph node metastasis of NVC; however, some older studies are dealing with this issue, e.g. PMID: 2297408. I suggest to mention in the introduction that there are no recent studies dealing with the incidence of neck metastasis in NVC and move the current text to the discussion.

3. The last paragraph of the “Injection technique” is not obvious. The brand information (GE Healthcare) refers for the Nanocoll and not for the “nuclear physicist”, I assume, therefore, it should stand there. Furthermore, there is a typo, as it is written as “nuclear physicistx”.

4. The statement “additional selective neck node dissection of level I-III of the involved side was performed” under “Clinical Consequences” is not correct, as in most of the cases bilateral ND was performed.

5. It is not obvious why so many (up to 10) sentinel nodes were removed. In my opinion, it is not necessary, as it does not improve the sensitivity of the procedure, just makes more chance of complications. Please, explain why so many nodes were harvested.

6. The discussion does not contain any literature comparison or overview. For this reason, I suggest to move the information from the introduction to the discussion with some more explanation of the importance of the SLNB in nasal NVC.

7. In my opinion, the suggestion “We advocate for the inclusion of a tumor diameter, drawing parallels with other head and neck cancers such as the lip and oral cavity.” cannot be made, based on the experiences of these 10 patients.

Reviewer #2: The manuscript titled "Sentinel NOSE: prospective feasibility study on sentinel lymph node biopsy in bulky nasal vestibule cancer" investigates the feasibility of sentinel lymph node biopsy (SLNB) in bulky nasal vestibule carcinoma (NVC). The study is novel and addresses an under-researched area, but several methodological and statistical issues need to be addressed to strengthen the manuscript and improve the transparency of its conclusions. Several aspects—particularly the justification of the feasibility threshold, the lack of power calculation, and the handling of missing data—require clarification to improve transparency and scientific rigor. Explicit acknowledgment of the study’s exploratory nature and limitations will enhance its contribution to the field.

Major Comments

1. Feasibility Threshold and Lack of Power Calculation:

The primary endpoint was defined as the successful identification of sentinel lymph nodes (SLNs) in ≥70% of patients (7 out of 10). The study reports a 100% success rate, exceeding the predefined feasibility threshold. However, no power calculation was performed, and the basis for the 70% threshold remains unclear. Was it derived from clinical expectations, previous studies, or expert consensus?

Without a power calculation or formal statistical justification, it is difficult to assess whether the chosen sample size (n = 10) provides sufficient confidence in meeting this threshold. Please provide a clear justification for the 70% threshold, referencing clinical or methodological precedent. In addition, reporting confidence intervals (e.g., for the observed success rate of 70%) will better contextualize the results.

2. Learning Curve Consideration:

The feasibility threshold explicitly accounts for a learning curve, but the manuscript does not explain how this was factored into the analysis or whether earlier cases were systematically less successful. Please clarify how the learning curve was addressed in the study design and analysis and discuss whether the inclusion of early cases might underestimate the true success rate after the learning phase.

3. Sample Size and Generalizability:

While the study reports successful SLN identification in all 10 patients, the small sample size limits the generalizability of the findings and reduces statistical power for secondary outcomes. Please consider to explicitly acknowledge the limitations of the small sample size in the discussion and discuss how these findings might guide the design of future studies with larger cohorts.

4. Interpretation of Occult Metastasis Findings:

The study reports a 50% occult metastasis rate, which is relatively high but expected given the inclusion of bulky tumors. How the inclusion criteria (bulky tumors) may have influenced the observed metastasis rates? Could you compare the findings with other SLNB feasibility studies to provide context.

5. Statistical Analyses and Multiplicity:

The study employs descriptive statistics and does not account for multiplicity in secondary outcomes. While this is acceptable for a feasibility study, the lack of statistical rigor could lead to overinterpretation. Acknowledge the exploratory nature of the secondary analyses and the lack of multiplicity adjustments. Avoid overinterpreting trends in secondary outcomes without sufficient power.

Minor Comments

- Clarity in Reporting: Ensure tables and figures are self-explanatory and include sufficient annotations to aid interpretation. For example, clearly label anatomical locations of SLNs in relevant figures. How were pain scores and complications assessed and reported in this study?

- Bias in Patient Selection: Please discuss whether patient selection (e.g., bulky tumors, single-center design) may introduce selection bias and limit generalizability.

- Ethical Considerations: The study received ethical approval, but it would be useful to reiterate measures taken to ensure patient safety, particularly in a novel diagnostic procedure.

**Do you want your identity to be public for this peer review?** For information about this choice, including consent withdrawal, please see our Privacy Policy

Reviewer #1: No

Reviewer #2: No

---

## [Author Response · Author response to Decision Letter 1]

23 Apr 2025

The editor mentioned that the is a confidentiality notice on the original protocol file under editor remark #2.

However, I can not find this notice, and we have not consciously added it; the full study protocol is even one of the supplementary materials. Therefore, there is no confidentiality notice on the attached original protocol file; we fully agree on publication of the protocol under the Creative Commons Attribution (CC BY) 4.0 license.

---

Dear Editor,

I would like to thank you and the reviewers for the time and effort invested in reviewing our manuscript titled “Sentinel NOSE: prospective feasibility study on sentinel lymph node biopsy in bulky nasal vestibule cancer” (PONE-D-24-55841). We appreciate the thoughtful and constructive feedback provided.

We have carefully considered all the comments and suggestions provided by the reviewers. As a result, we have addressed each point in the text found below, and performed necessary revisions. We believe these changes have enhanced the clarity and overall quality of the manuscript, and we hope that the revised version meets the expectations of the reviewers.

We look forward to hearing your feedback and trust that the revised manuscript will be suitable for publication in PLOS One.

Sincerely,

On behalf of all co-authors,

Michal Czerwinski

Reviewer #1: The manuscript, titled “Sentinel NOSE: prospective feasibility study on sentinel lymph node biopsy in bulky nasal vestibule cancer” is a very well written paper, containing important information on a clinically relevant topic. In my opinion, the study should be published; however, before publishing some minor revisions should be applied.

1. The cut-off of 1.5 cm for the indication of SLNB is not sufficiently explained.

The rationale for this cut-off is based on previous institutional- and multicenter retrospective research and explained in the introduction:

“Our previous research identified a subset of patients with bulky tumors (≥1.5 cm in diameter and/or ≥2 cm³ in volume) facing a high nodal relapse risk of up to 40%. Given that the presence of even a single nodal metastasis decreases survival in head-and-neck cancers (HNC) by approximately 50, it raises consideration for elective neck treatment in bulky disease, either through surgery or irradiation.”

2. In the first paragraph of the introduction, the authors refer to studies which describe the nodal recurrence of NVC; however, the present study investigate how to improve accurate primary staging. I understand that there are no recent studies available on the incidence of occult lymph node metastasis of NVC; however, some older studies are dealing with this issue, e.g. PMID: 2297408. I suggest to mention in the introduction that there are no recent studies dealing with the incidence of neck metastasis in NVC and move the current text to the discussion.

The rationale for the parallel between nodal recurrences and occult metastases is based on observed isolated nodal recurrences after thorough radiological (and often cytological) staging, often manifesting after a short period of time (months). This strongly supports the assumption that these (occult) metastases were present at diagnosis, but were not detected due to the limitations of current staging techniques. The sentinel node procedure is proposed as an improved staging method to detect these occult metastases and reduce the incidence of nodal recurrences.

Therefore, we find it important to keep this rationale in the introduction.

3. The last paragraph of the “Injection technique” is not obvious. The brand information (GE Healthcare) refers for the Nanocoll and not for the “nuclear physicist”, I assume, therefore, it should stand there. Furthermore, there is a typo, as it is written as “nuclear physicistx”.

This paragraph has been amended based on your recommendation.

4. The statement “additional selective neck node dissection of level I-III of the involved side was performed” under “Clinical Consequences” is not correct, as in most of the cases bilateral ND was performed.

This paragraph has been amended based on your recommendation to improve clarity:

“additional selective neck node dissection of level I-III of the involved side was performed of either the involved side in case of a lateralized tumor, or bilaterally in case of a midline tumor”

5. It is not obvious why so many (up to 10) sentinel nodes were removed. In my opinion, it is not necessary, as it does not improve the sensitivity of the procedure, just makes more chance of complications. Please, explain why so many nodes were harvested.

Since previous research on the lymph drainage patterns of nasal vestibule cancer is limited and because this was an exploratory study, we did not enforce restrictions on the number of nodes to be removed.

The highest number of harvested nodes in a single patient was six, and as you can see in S2 table, five out of the six harvested nodes were found in a single level (IB); these nodes were hot and all possibly containing metastases and were therefore removed. In most other patients, only one to four nodes were removed.

6. The discussion does not contain any literature comparison or overview. For this reason, I suggest to move the information from the introduction to the discussion with some more explanation of the importance of the SLNB in nasal NVC.

The discussion contains comparisons with other studies on the sentinel node procedure in head-and-neck cancers, both on success rates and encountered challenges and also previous retrospective research. However, we agree that the discussion was lacking in parallels to previous research on nodal recurrence rates in NVC; therefore, the following paragraph has been moved to the discussion section:

“Notably, elective neck treatment is generally not advocated in ES-NVC [8], as literature reports overall nodal recurrence rates ranging from 0% to 29% in ES-NVC [4-7]; therefore the observed trend of high incidence of occult metastases appears to be specific for bulky NVC.”

7. In my opinion, the suggestion “We advocate for the inclusion of a tumor diameter, drawing parallels with other head and neck cancers such as the lip and oral cavity.” cannot be made, based on the experiences of these 10 patients.

We agree; the statement has been amended to “We advocate to further investigate the prognostic relevance of tumor size, drawing parallels with other head and neck cancers such as the lip and oral cavity.”

Reviewer #2: The manuscript titled "Sentinel NOSE: prospective feasibility study on sentinel lymph node biopsy in bulky nasal vestibule cancer" investigates the feasibility of sentinel lymph node biopsy (SLNB) in bulky nasal vestibule carcinoma (NVC). The study is novel and addresses an under-researched area, but several methodological and statistical issues need to be addressed to strengthen the manuscript and improve the transparency of its conclusions. Several aspects—particularly the justification of the feasibility threshold, the lack of power calculation, and the handling of missing data—require clarification to improve transparency and scientific rigor. Explicit acknowledgment of the study’s exploratory nature and limitations will enhance its contribution to the field.

Major Comments

1. Feasibility Threshold and Lack of Power Calculation:

The primary endpoint was defined as the successful identification of sentinel lymph nodes (SLNs) in ≥70% of patients (7 out of 10). The study reports a 100% success rate, exceeding the predefined feasibility threshold. However, no power calculation was performed, and the basis for the 70% threshold remains unclear. Was it derived from clinical expectations, previous studies, or expert consensus?

Without a power calculation or formal statistical justification, it is difficult to assess whether the chosen sample size (n = 10) provides sufficient confidence in meeting this threshold. Please provide a clear justification for the 70% threshold, referencing clinical or methodological precedent. In addition, reporting confidence intervals (e.g., for the observed success rate of 70%) will better contextualize the results.

We understand the critique on the absence of a power calculation very well and have discussed this thoroughly before the start of the study. Due to the descriptive and explorative nature of this study and endpoints unrelated to clinical outcomes such as disease control or survival, a power calculation has not been performed.

Furthermore, since NVC is a rare cancer with an estimated incidence of 50-60 patients/year in the entirety of the Netherlands, this imposed a practical limit on a feasible number of study participants in a single center, in a limited amount of time. Especially because we aimed to include only patients with large tumors for which we demonstrated in previous studies that they have a relatively high risk of occult metastases.

(We have added this comment in the discussion section of the manuscript)

This work was meant to serve as a feasibility study to a follow-up multi-center investigation, which will explore the clinical added value of the sentinel node procedure. By expert consensus, the research team holds the view that an invasive diagnostic procedure must demonstrate a substantial likelihood of success and an acceptable level of burden for the patient before being evaluated in subsequent research. Clinical practice indicates that in subsites where the sentinel node procedure is part of standard diagnostics, success rates are typically very high (90-100%). For this target group as well, the ultimate goal would ideally be a >90% success rate. However, it was essential to account for an initial learning curve.

2. Learning Curve Consideration:

The feasibility threshold explicitly accounts for a learning curve, but the manuscript does not explain how this was factored into the analysis or whether earlier cases were systematically less successful. Please clarify how the learning curve was addressed in the study design and analysis and discuss whether the inclusion of early cases might underestimate the true success rate after the learning phase.

Despite the 100% success rate, previous concerns on an initial learning curve manifested in the first patient, where a high number of potential sentinel nodes was identified on imaging probably due to tracer leakage in the esophagus. We added the following passage in the discussion section to underline this issue:

“Despite extensive experience with SLNB for oral tumors at our center, a learning curve was observed for this new subsite. Particularly in the first patient, nasopharyngeal contamination resulting in a very high number of sentinel nodes on imaging highlighted the need for proficiency and carefulness in injecting.”

3. Sample Size and Generalizability:

While the study reports successful SLN identification in all 10 patients, the small sample size limits the generalizability of the findings and reduces statistical power for secondary outcomes. Please consider to explicitly acknowledge the limitations of the small sample size in the discussion and discuss how these findings might guide the design of future studies with larger cohorts.

More emphasis has been put in the discussion and conclusion on the small sample size and limited generalizability of the results, advocating for further research:

“However, this positive result still has to be interpreted with caution due to the small sample size of ten.”

“There are clear limitations to the study. The small size of the cohort, the limited follow-up and the study’s exploratory nature leaves relevant questions open, such as the generalizability of these results and ultimate regional control rate after SLNB.”

“Further research is necessary to confirm these findings and investigate the efficacy of SLNB in reducing morbidity and improving oncological outcomes in bulky NVC.”

4. Interpretation of Occult Metastasis Findings:

The study reports a 50% occult metastasis rate, which is relatively high but expected given the inclusion of bulky tumors. How the inclusion criteria (bulky tumors) may have influenced the observed metastasis rates? Could you compare the findings with other SLNB feasibility studies to provide context.

The reviewer is correct in that the observed high incidence of occult metastasis is because we included bulky tumors, as the incidence of nodal recurrences in cN0 NVC overall is lower (10-15%); therefore, specifically patients with bulky NVC were selected for the sentinel lymph node procedure as previous research suggested that they might be at high risk of nodal recurrence and therefore occult metastases as mentioned in the introduction:

“Our previous research identified a subset of patients with bulky tumors (≥1.5 cm in diameter and/or ≥2 cm³ in volume) facing a high nodal relapse risk of up to 40% [3, 9]. Given that the presence of even a single nodal metastasis decreases survival in head-and-neck cancers (HNC) by approximately 50% [2], it raises consideration for elective neck treatment in bulky disease, either through surgery or irradiation [10, 11]. However, in the era of minimally invasive treatments, a novel approach has bridged the gap between imaging and elective treatment of the neck.”

The fact that our study concerns specifically bulky tumors has been emphasized more in the discussion, and an comparison to the incidence of occult metastases in other mucosal head and neck cancers has been added:

“Although no definitive conclusions can be drawn from this small feasibility study, interestingly, the remarkable incidence of occult metastases in 50% of patients with bulky NVC corroborates previous findings in retrospective cohorts, particularly the 40% nodal recurrence risk in bulky disease [3, 9]. Notably, elective neck treatment is generally not advocated in ES-NVC [8], as literature reports overall nodal recurrence rates ranging from 0% to 29% in ES-NVC [4-7]; therefore the observed trend of high incidence of occult metastases appears to be specific for bulky NVC. In context, a recent large meta-analysis on diagnostic accuracy of SLNB in mucosal head-and-neck cancers found a pooled prevalence of clinically occult metastases of 30% in oropharyngeal- and laryngeal/hypopharyngeal cancer [15]. However, this analysis included T1-T4 tumors without distinguishing between bulky and non-bulky disease, limiting direct comparisons.”

5. Statistical Analyses and Multiplicity:

The study employs descriptive statistics and does not account for multiplicity in secondary outcomes. While this is acceptable for a feasibility study, the lack of statistical rigor could lead to overinterpretation. Acknowledge the exploratory nature of the secondary analyses and the lack of multiplicity adjustments. Avoid overinterpreting trends in secondary outcomes without sufficient power.

We agree, and added the following passage to the passage mentioning the incidence of occult nodal metastases:

“It is also important to note that the current study was not powered to investigate this outcome thoroughly, and as such, these findings should be interpreted only as an observed trend.”

Minor Comments

- Clarity in Reporting: Ensure tables and figures are self-explanatory and include sufficient annotations to aid interpretation. For example, clearly label anatomical locations of SLNs in relevant figures. How were pain scores and complications assessed and reported in this study?

- Bias in Patient Selection: Please discuss whether patient selection (e.g., bulky tumors, single-center design) may introduce selection bias and limit generalizability.

- Ethical Considerations: The study received ethical approval, but it would be useful to reiterate measures taken to ensure patient safety, particularly in a novel diagnostic procedure.

Efforts have been made to clarify the figures and tables and emphasize patient selection bias (particularly bulky tumors). Regarding patient safety, as additional measures, strict monitoring of pain levels was performed and the study protocol specified that retropharyngeal, buccal or pre-auricular and level III-VI SLNs would not be surgically removed if higher echelon sentinel lymph nodes were present following a consensus decision, due to the height

---

## [Decision Letter · Decision Letter 1]

Sentinel NOSE: prospective feasibility study on sentinel lymph node biopsy in bulky nasal vestibule cancer

PONE-D-24-55841R1

Dear Dr. Czerwinski,

We’re pleased to inform you that your manuscript has been judged scientifically suitable for publication and will be formally accepted for publication once it meets all outstanding technical requirements.

Kind regards,

Rong-San Jiang

Academic Editor

PLOS ONE

Additional Editor Comments (optional):

Reviewers' comments:

Reviewer's Responses to Questions

**Comments to the Author**

Reviewer #1: All comments have been addressed

Reviewer #2: All comments have been addressed

2. Is the manuscript technically sound, and do the data support the conclusions?

Reviewer #1: Yes

Reviewer #2: (No Response)

3. Has the statistical analysis been performed appropriately and rigorously?

Reviewer #1: Yes

Reviewer #2: (No Response)

4. Have the authors made all data underlying the findings in their manuscript fully available?

Reviewer #1: Yes

Reviewer #2: (No Response)

5. Is the manuscript presented in an intelligible fashion and written in standard English?

Reviewer #1: Yes

Reviewer #2: (No Response)

Reviewer #1: The authors have accordingly revised the manuscript. The manuscript is suitable for publication in my opinion.

Reviewer #2: (No Response)

**Do you want your identity to be public for this peer review?** For information about this choice, including consent withdrawal, please see our Privacy Policy

Reviewer #1: No

Reviewer #2: No

---

## [Editor Report · Acceptance letter]

PONE-D-24-55841R1

PLOS ONE

Dear Dr. Czerwinski,

I'm pleased to inform you that your manuscript has been deemed suitable for publication in PLOS ONE. Congratulations! Your manuscript is now being handed over to our production team.

Kind regards,

on behalf of

Dr. Rong-San Jiang

Academic Editor

PLOS ONE